# Comparison of Selected Characteristics of Slovak and Polish Representatives in Kickboxing

**DOI:** 10.3390/ijerph191710507

**Published:** 2022-08-23

**Authors:** Pavel Ruzbarsky, Kristina Nema, Marek Kokinda, Łukasz Rydzik, Tadeusz Ambroży

**Affiliations:** 1Department of Sports Kinanthropology, Faculty of Sports, University of Presov, 080-01 Presov, Slovakia; 2Institute of Sports Sciences, University of Physical Education in Krakow, 31-541 Krakow, Poland

**Keywords:** combat sports, female athletes, technical and tactical indicators, physical fitness

## Abstract

Background: Previous kickboxing studies conducted over the last three decades have focused on improving performance through physical fitness characteristics; however, most of the research focused on men. The objective of this study is to assess the level of physical fitness of Slovak and Polish female kickboxers in the highest sport level as well as to compare the differences between them. Methods: The study included 20 female kickboxers on the highest level of sport performance, 10 from Slovakia (body mass—53.59; height—166.45) and 10 from Poland (body mass—60.35; height —169.95), from senior categories, with a mean age of 23 ± 2. The selection criteria included training experience and sports level. The level of physical fitness was evaluated by the following tests: Cooper’s test, 50 m sprint, sit-ups, flexed arm hang and standing long jump. Technical and tactical indicators were used to assess the sports skill level during the competition. Results: The test results of both groups are very comparable based on the evaluated data of Kruskal–Wallis ANOVA. The attack activity index together with sit-ups demonstrated statistically significant differences (*p* < 0.05) between Slovak and Polish kickboxers. A strong positive correlation was proven between technical and tactical indicators and physical fitness tests in the results of both groups. Conclusions: There exists a positive correlation between the technical and tactical indicators of kickboxers and the explosive strength of lower limbs, strength endurance, abdominal muscle strength, speed and aerobic endurance. Differences between Slovak and Polish kickboxers have been shown in the attack activity index and in the level of strength endurance. The level of speed and explosive power of the lower limbs was comparable in both groups of kickboxers.

## 1. Introduction

Kickboxing is a modern combat sport that is currently very popular and has become one of the pre-Olympic sports. A kickboxing match consists of three rounds in the senior categories lasting 2 min with a one-minute break between rounds, with each match having a time limit of 8 min [1]. The sport is growing in popularity on the world stage by count. According to the Word Association of Kickboxing Organizations (WAKO), kickboxing is represented by 130 countries from five continents, of which 105 are officially recognized by the International Olympic Committee [2]. Success in kickboxing is associated with psychological [3], physiological [4,5], and biomechanical [6] aspects.

Although the majority of mixed martial artists are male, there are also female athletes who train and compete in this male-dominated combat sport. Male and female who are engaged in combat sports have different needs to address regarding physical fitness and athletic performance. Although men and women increase strength at the same rate, performance can be enhanced by the application of a female-specific training program that addresses physical limitations while being contoured to the combat sports [7].

The optimal level of physical fitness of a competitor is the key element of efficiency in a sport competition [8]. A kickboxing fight is based on punching and kicking techniques, with fighters allowed to strike only specified parts of the opponent’s body. The repertoire of the techniques requires adequate physical fitness to use them effectively [9]. Studies involving simulated and official contest suggested that kickboxing is strongly anaerobic, highly stressing the cardiovascular system [10] and demanding body power to perform offensive and defensive actions and specific foot movements at the required frequency [11]. Physical effort is based on sub-maximal and maximal loads. The physiological profiles of competitors show that the physical training of kickboxing should by aimed at increasing both aerobic and anaerobic capacity [8].

Kickboxers were observed to have high levels of strength, power, aerobic and anaerobic capacity combined with technical and tactical skills [12]. The strength of the upper limbs in kickboxing has been evaluated by measuring flexed arm hang and grip strength [8,13]. The strength and dynamics of lower limbs were evaluated by measuring the distance of a jump [14,15], and for evaluation of abdominal strength, it was sit-ups [8,16]. For measurement of an athlete’s aerobic endurance level, we used the Cooper test [16]. Speed was evaluated by 50 m sprint [8].

Previous kickboxing studies conducted over the last three decades have focused on improving performance through physical fitness characteristics; however, most of the research focused on men [5,8,17,18,19,20], and only a small part of the studies included women in the research [4,21]. The Polish Senior Team, which represents kickboxing in all formulas in the Polish Kickboxing Association, has 87 females [22], while in Slovakia, the number of female kickboxers representing the Slovak Kickboxing Federation [23] in international competition is lower, as it is only 12 females. At the last world championships, the average age in the female’s competition was 25 years. Despite keeping start-up statistics, a considerable deficit of research regarding the level of training and physical fitness of female competitors was noted.

The objective of this study is to assess the level of physical fitness of Slovak and Polish female kickboxers in highest sport level as well as to verify the fitness profiles between neighboring countries with different results in kickboxing in the world arena results.

## 2. Materials and Methods

### 2.1. Participants

The study included 20 female kickboxers, 10 from Slovakia and 10 from Poland, who were preparing for the competitions at the international level in senior categories, with a mean age of 23 ± 2. They had fought an average of 15 matches a year in national and international competitions, and their training experience depending on their age ranged from 8 to 10 years. The frequency of training sessions was 5–6 times a week, 1.5 h each.

Inclusion criteria for the study were: a minimum of 5 years of training experience, a positive recommendation from the coach, good health, active competitions, success at world and international levels, and gender. The exclusion criteria were a training length of less than 5 years, injuries and poor health, no competition achievements, and male gender. The athletes were measured for body mass, body height and physical fitness. The anthropometric characteristics of the participants are presented in Table 1.

### 2.2. Physical Fitness Tests

The physical fitness of the participants was assessed by a selected test taken from the developed ICSPFT (International Committee on the Standardization of Physical Fitness Tests) tests and EUROFIT (European Fitness Test) [24]. The choice of tests was conditioned by previous research carried out in this area [8,13]. The assessment of physical fitness includes the following tests:Cooper’s test—12 min run, distance is measured.50 m sprint—Participants run a distance of 50 m as quick as they could.Sit-ups—Evaluating abdominal muscle strength, participants did as many sit-ups as they could in 30 s. (The subject lies on the mattress, their feet are 30 cm apart and their knees bent at 90 degrees, with hands on their neck. A partner holds the subject’s feet so they stay on the ground. On a signal, the subject performs sit-ups touching their knees with their elbows coming back to lying down.) The test lasts 30 s.Flexed arm hang—Evaluating shoulder girdle strength. The subject takes a position with the armed flexed and chin clearing the bar and holds this position for as long as possible. The total time in seconds is evaluated.Standing long jump—Jumping with both feet from standing. The test measures the distance jumped in cm, which is an indicator of the possibility to quickly create strength. Participants jumped from a standing position as far as they could. (The subject stands with their feet lightly spread behind the start line, they bend their knees and move their arms backward; then, they move their arms forward, bounce their feet from the ground and make a jump as long as possible. They land on both feet in a standing position. The test is taken twice. The longer jump is recorded and rounded to the nearest cm.)

The tests were supervised by the authors. Tests were performed at the beginning of the training camp. Tests 2, 4, and 5 were performed on the first day, whereas tests 1 and 3 were performed on the second day. Two days before the tests, training intensity was reduced to 30–40% of the baseline levels.

### 2.3. Measuring the Indicators of Technical and Tactical Training

To verify the sports skill level during the tournament, we performed the analysis of the fight and made relevant calculations. The observations were carried out by two experts with kickboxing coaching qualifications. The researchers recorded the recording on special spreadsheets. The results were then added up, and the average of the two records was taken. The analysis of each round was based on the digital recording of a fight. Then, the indicators of technical and tactical training were computed using the following formulas [8,13,25].

Efficiency of the attack (*S_a_*)
$$S_{a}=\frac{n}{N}$$

*N*—Number of bouts.

*n*—Number of attacks awarded 1 pt. 

* In kickboxing, each fair hit is awarded 1 pt.

Effectiveness of the attack (*E_a_*)
$$E_{a}=\frac{number\;of\;efective\;attacks}{number\;of\;all\;attacks}\times 100$$

∗ An effective attack is a technical action awarded a point.∗ Number of all attacks is the number of all offensive actions.

Activeness of the attack (*A_a_*)
$$A_{a}=\frac{number\;of\;all\;registered\;offensive\;actions\;of\;a\;kickboxer}{number\;of\;bouts\;fought\;by\;a\;kickboxer}$$

### 2.4. Bioethical Committee

The study was conducted according the guidelines of the declaration of Helsinki (2008). Obtaining the participant’s written consent was the precondition for their participation in the project. The research was approved by the Bioethics Committee at the Regional Medical Chamber (No. 287/KBL/OIL/2020).

### 2.5. Statistical Analysis

Statistical analysis of the data was conducted with the use of Statistica 13.5. Statistical analysis was performed using nonparametric tests due to the small number of research groups. Violation of normality and verification of the appropriate choice of nonparametric methods were verified by the Shapiro–Wilk test. From descriptive statistics, the median was chosen from the measures of the central tendency, and the quartile deviation was chosen from the measures of variability. The Kruskal–Wallis ANOVA was chosen to assess the significance of the differences between Slovak and Polish female kickboxers. Canonical correlation was used to determine the relationship between fitness tests and indicators of technical and tactical training. The level of statistical significance was set to *p* < 0.05. Substantive significance was assessed according to Cohen [26]: 0.1 < r < 0.3 (small), 0.3 < r < 0.5 (medium), r > 0.5 (high).

## 3. Results

Selected characteristics of physical fitness were compared between Slovak and Polish female kickboxers. The results of fitness tests are shown in Table 2.

From the point of view of average values, we can state that with the exception of the sit-ups test, the Slovak kickboxers achieve higher performance. At the same time, the group of Slovak kickboxers also showed higher variability of performances in the following tests: Cooper’s test, sit-ups and flexed arm hang. On the contrary, the Polish kickboxers showed higher performance variability in the linear speed test (50 m sprint) and standing long jump test. The recorded differences between the groups were minimal in the performed tests; in recalculation, the differentiation ranged from 2 to 5%.

Results of the indicators of technical and tactical training are presented in Table 3. Based on the results of the mean values, we can state that with the exception of the efficiency activity index, Slovak kickboxers showed higher values.

The test results of both groups are very comparable based on the evaluated data of Kruskal–Wallis ANOVA (Table 4). The attack activity index together with sit-ups demonstrated statistically significant differences (*p* < 0.05) between Slovak and Polish kickboxers. Effect size evaluated according to Cohen [26] was rated as medium (0.3 < r < 0.5) in the attack efficiency index. The effect size of the attack effectiveness index was rated as small (0.1 < r < 0.3). In the Cooper test, sit-ups, flexed arm hang and standing long jump, the effect size was rated as small (0.1 < r < 0.3). In the 50 m sprint test, the effect size was rated as medium (0.3 < r < 0.5).

A strong positive correlation was proven between technical and tactical indicators and physical fitness test for the results of Slovak and polish female kickboxers. Correlation was not statistically significant (Table 5).

## 4. Discussion

Kickboxing athletes require muscle strength, strength endurance, speed and aerobic endurance to effectively perform and sustain the technical and tactical actions in the fight including kicking, punching, blocking and pushing [21,25,27]. The activeness, effectiveness and efficiency of the attack depend on the level of muscle strength of the upper, lower and middle part of the body [8]. The results shown that there exists a positive correlation in the research group of females between technical and tactical indicators of kickboxers and the explosive strength of lower limbs, strength endurance, abdominal muscle strength, speed and aerobic endurance, which is in agreement with the study results from Rydzik and Amborży [8]. The attack activity index demonstrated statistically significant differences between Slovak and Polish kickboxers. Slovak kickboxers achieved better values of the attack activity index, which relates to the number of the techniques performed during each round, which has a strong statistically significant, positive correlation with aerobic endurance according to Rydzik and Amborży [8]. This statements agree with the results of our research when Slovak kickboxers achieved a higher level of aerobic endurance in Cooper’s test.

Developing the proper timing in kickboxing fights requires a certain level of speed, reaction time and understanding the various sparring situations such as counter-act and scoring in a fight [21,28]. The level of speed (action implementation) was comparable in both groups of kickboxers, although the Polish kickboxers achieved better results, but no significant differences were found. Similar results in terms of speed were achieved by female taekwondo athletes at the international level with values of 7.21 ± 0.45 s [29] and 7.6 s [30].

Significant differences between Slovak and Polish kickboxers were found in the level of strength endurance in favor of Slovaks. In other combat sports, strength endurance as assessed by the sit-ups test was lower in taekwondo female athletes, 54.20 ± 6.84 [31] and 55.2 ± 6.1 [32], compared to Slovak and Polish female kickboxers. We did not find statistically significant differences between the Slovak and Polish female kickboxers in the level of strength endurance of the upper limbs. The level of strength endurance is comparable in both groups. However, in comparison with female karate athletes at the international level, both Slovak and Polish kickboxers had a significantly higher level of strength endurance when female karate athletes achieved results of 30.42 ± 14.43 s in the flexed arm hang test [33]. Overall, the isometric strength of the upper limbs is particularly important for achieving high-level amateur and elite kickboxing performance [21].

In kickboxing [10] as in other combat sports such as taekwondo [34] and karate [35], it has been reported that kicks are used a lot during the fight, and it is thought that these kicks need to be thrown at a certain speed in order to be accurate; therefore, explosive power of the lower limbs must be high in order to be successful in such sports [36,37]. The level of explosive power of the lower limbs was also comparable to the value of the standing long jump test, which we did not to be find statistically significant between Slovak and Polish kickboxers, but Polish kickboxers achieved better results. Compared to female fighters at the international level in various martial arts (taekwondo, karate), Slovak and Polish kickboxers had a better level of explosive power in the lower limbs than taekwondo athletes (192.47 ± 14.25 cm) [31] but worse than karate women (215.08 ± 15.01 cm) [33].

### Limitation of the Study

Some limitations in current study were observed. It is worth noting that the study was conducted on a group of elite female kickboxers, which additionally limits the sample size due to the presented sports skill level. A group of 10 women from each country is not able to show the details of training for the population. Therefore, the present study is illustrative. The statistical analysis focused on the evaluation of technical–tactical training indicators from the entire bout. We did not verify the indices in individual rounds of combat, which is also a limitation of our study. In addition, the indices were calculated on the basis of real duels (not simulated fights), so we did not have the opportunity to verify fatigue by measuring lactate concentration or pulse. In addition, by studying elite groups from two countries, we were not able to compare the same weight categories, as not all of them included female representatives.

## 5. Conclusions

It is well known that understanding the characteristics of athletes on the highest level can provide useful information regarding what is truly needed for success in competition. This study assesses the level of physical fitness of Slovak and Polish female kickboxers in the highest sport level as well as compares the differences between them. The results show that there exists a positive correlation between technical and tactical indicators of kickboxers and the explosive strength of lower limbs, strength endurance, abdominal muscle strength, speed and aerobic endurance. The attack activity index demonstrated statistically significant differences between Slovak and Polish kickboxers in the research group. Significant differences between Slovak and Polish kickboxers in the research group were found in the level of strength endurance in favor of Slovaks. The level of speed and explosive power of the lower limbs was comparable in both study groups of kickboxers. Poland has a larger membership base, which creates more female kickboxer rivals in each category, which can also have an effect regarding the higher performance level.

### Practical Implication

The present study may indicate a modification of the training process in female kickboxing through the use of the all-round development of female athletes, particularly in the aspects of endurance, speed and strength, which may influence the improvement of technical–tactical activities. However, due to the small sample size, there is a need to conduct a larger study on a larger population.

## Figures and Tables

**Table 1 ijerph-19-10507-t001:** Anthropometric characteristics.

	Slovak Kickboxers (n = 10)		Polish Kickboxers (n = 10)	
	Median	Quartile Range	Median	Quartile Range
Body mass (kg)	53.59	4.80	60.35	4.70
Height (cm)	166.45	3.53	169.95	3.00

**Table 2 ijerph-19-10507-t002:** Physical fitness.

Variables	Slovak Kickboxers (n = 10)		Polish Kickboxers (n = 10)	
	Median	Quartile Range	Median	Quartile Range
Cooper’s test (m)	2445.00	550.00	2385.00	330.00
50 m sprint (s)	7.65	0.45	7.85	0.70
Sit-ups (n)	67.00	12.00	64.00	7.00
Flexed arm hang (s)	60.00	30.00	57.00	12.00
Standing long jump (m)	203.50	9.00	212.00	28.00

**Table 3 ijerph-19-10507-t003:** Activeness, effectiveness and efficiency of the attacks.

Variables	Slovak Kickboxers (n = 10)		Polish Kickboxers (n = 10)	
	Median	Quartile Range	Median	Quartile Range
Activeness	54.67	18.34	44.58	19.33
Effectiveness	9.95	5.21	9.49	4.61
Efficiency	3.66	4.16	5.67	4.67

**Table 4 ijerph-19-10507-t004:** Kruskal–Wallis ANOVA between Slovak and Polish kickboxers.

Variables	Kruskal–Wallis ANOVA		Effect Size
	H	*p*	*r*
Activeness	4.81	0.03	0.48
Effectiveness	0.05	0.82	0.04
Efficiency	1.96	0.16	0.30
Cooper’s test	0.57	0.44	0.24
50 m sprint	0.70	0.40	0.34
Sit-ups	14.30	0.01	0.10
Flexed arm hang	3.20	0.07	0.18
Standing long jump	2.41	0.12	0.16

Legend: H—value of Kruskal–Wallis ANOVA, *p*—significance level.

**Table 5 ijerph-19-10507-t005:** Canonical correlation.

**Slovak Kickboxers**							
	Correlation	Eigenvalue	Wilks Statistics	F	Num D.F	Denom D.F	Sig.
1	0.87	12.904	0.036	0.922	15.000	5.923	0.586
2	0.67	0.809	0.500	0.311	8.000	6.000	0.931
3	0.31	0.105	0.905	0.140	3.000	4.000	0.931
**Polish Kickboxers**							
1	0.966	13.982	0.012	1.548	15.000	5.923	0.309
2	0.874	0.874	0.874	0.874	0.874	0.874	0.874
3	0.467	0.467	0.467	0.467	0.467	0.467	0.467

## Data Availability

The data presented in this study are available on request from the corresponding author.

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
