# Peer review of "Comparison of Selected Characteristics of Slovak and Polish Representatives in Kickboxing"

_ijerph, 2022, doi:10.3390/ijerph191710507_

Round 1
Reviewer 1 Report
Dear authors,
Thank you for writing this interesting paper. It is a novel study with a unique sample, but it would be beneficial to the reader if you could kindly address the following corrections and suggestions:
Abstract
- Semi-colons between sections should be replaced with full stops / periods.
- "0,05" should be "0.05" (with the decimal point)
Line 26: unnecessary capitalisation of "the"
Introduction
- What is the average age of a competitive / elite female kickboxer? Is it younger than in other combat sports? The sample ranging from 21 to 25 is quite young.
- Are there any statistics on the numbers of female kickboxers in Poland and Slovakia? Are they within national governing bodies?
- How did you access this population in the first place?
- Are any of the authors involved in kickboxing or in the preparation of these athletes for their competitions?
- Why was this study conducted? Do Slovak and Polish kickboxers follow different strength training routines? Is one of the nations more dominant in the sport than the other one?
Methods
What was the minimum amount of experience required to be included in the sample?
p. 4: Were there any specific ethical measures that you followed / considered.
Table / results
The height and weight differences between the Slovak and Polish kickboxers is interesting. Why might this be? Just by chance? Or is there a general difference in body size and mass between the national populations?
This article is a short one considering that IJERPH is an online journal with a possibility for long, detailed papers. As such, I would suggest taking up more space with details on the context for the study, the exact procedures and an analysis and interpretation that considers the broader national populations in question.
Thanks in advance for your efforts.
Author Response
Abstract
1. Semi-colons between sections should be replaced with full stops / periods and "0,05" should be "0.05" (with the decimal point)
Modified according to the reviewer’s recommendation
2. Line 26: unnecessary capitalisation of "the"
Modified according to the reviewer’s recommendation
Introduction
3. What is the average age of a competitive / elite female kickboxer? Is it younger than in other combat sports? The sample ranging from 21 to 25 is quite young.
The average competitive age of elite female kicboxers in senior categories was on last world senior championship 25 years. But also in other combat sports the average age of female fighter is lower than man. It can by caused by the end of studies at university and starting work, or pregnancy and starting a family.
4. Are there any statistics on the numbers of female kickboxers in Poland and Slovakia? Are they within national governing bodies?
Yes, there are, Slovak kickboxing union has a statistics on the numbers of female kickboxers. On international level Slovakia has a 12 female compete on senior categories.
5. How did you access this population in the first place?
Two of author (one from Slovakia and one from Poland) prepare kickboxers from our study on competitions and cooperate with national union on physical fitness testing.
6. Are any of the authors involved in kickboxing or in the preparation of these athletes for their competitions?
Yes, one of author was female kickboxer on international level and has several medals from world cups. And two of authors cooperate with national union and prepare kickboxers from our study on competitions as we mention up.
7. Why was this study conducted? Do Slovak and Polish kickboxers follow different strength training routines? Is one of the nations more dominant in the sport than the other one?
Because we want to assess the level of physical fitness of Slovak and Polish female kickboxers in highest sport level as well as to compare the differences between them. Reason was that are differences in sport preparation between Slovak and Polish. The fighting style of polish kickboxers is rather offensive compared to Slovak kickboxers. Also in other countries there are typical fighting style, or fighting ,,school“ which is characterized for them. It’s hard to say that one of the nations is more dominant in this sport, but according the ranking of wako (word kickboxing union), Poland has better results because Poland has 8 place and Slovakia is on the 21 place in this ranking.
Methods
8. What was the minimum amount of experience required to be included in the sample?
Minimum amount of experience required to be included in the sample was 5 years, but the sport level was decisive.
9. Were there any specific ethical measures that you followed / considered.
No, just what was mentioned in part 2.4 Bioethical committee
Table / results
10. The height and weight differences between the Slovak and Polish kickboxers is interesting. Why might this be? Just by chance? Or is there a general difference in body size and mass between the national populations?
Somatic predisposition is one of the determining factors of success in a fight. According to the weight categories, the competition program is planned, and the height requirements, together with the span of the upper and lower limbs, form the basis of the tactics in the fight. From the point of view of the investigated nationalities, it is possible to state a comparable somatotypology. The differences between the research subjects were random as this was a small sample size.
Reviewer 2 Report
I want to congratulate the authors for their work and decation. This was reviewed with interest. I have a few questions regarding the manuscript along with some comments to hopefully help create a more succinct version for the readers of the journal:
- Material and Methods:
- How many observers reviewed the recordings to establish the technical and tactical indicators? Were any of them external to the researchers? (line 110).
- Do you believe that using an isometric fatigue method is the most efficient way to measure strength in a sport like kickboxing?
- Results: The analysis of the correlation between technical and tactical indicators is very poor. How are these successes or errors distributed over the combat time? What influence does fatigue have on this aspect? A deeper analysis of this area is recommended.
- Discussion: Establishing this type of correlation in a phenomenon as complex as the one that arises in this sport can lead to incorrect analyses. I believe that the authors should carry out a more detailed analysis (lateralization, upper-lower limb, time within each round, pre-fight fatigue levels, etc.), in order to draw more powerful conclusions.
- Given the difference in weight of the participants, together with the small sample size, to what extent are the data obtained generalizable to all categories of the sport?
- Conclusion:
- Based on the previous questions, do the authors believe that the data obtained can be extrapolated to the generality of women practicing this sport at a high level?
Author Response
Material and Methods
1. How many observers reviewed the recordings to establish the technical and tactical indicators? Were any of them external to the researchers? (line 110).
There were two observers reviewed the recordings to establish the technical and tactical indicators, one from Slovakia and one from Poland.
2. Do you believe that using an isometric fatigue method is the most efficient way to measure strength in a sport like kickboxing?
Isometric testing has its justification from the point of view of prevention and assessment of the state of the support-motion system of athletes. When flexed arm hang, it is crucial to stabilize the area of the trunk, shoulder blades and to optimize breathing in relation to the time interval. For kickboxing, the strength of the torso is decisive for the execution of rotational movements, and in terms of prevention, the scapula plays a key role in stabilizing the shoulder joint.
Results
3. The analysis of the correlation between technical and tactical indicators is very poor. How are these successes or errors distributed over the combat time? What influence does fatigue have on this aspect? A deeper analysis of this area is recommended.
From the correlation analysis, it is possible to proceed to more extensive processing. From the point of view of the state of the support-movement system, it is crucial to optimize its activity, which creates a determining factor for success in the fight. The mutual relationship of general and specific sport performance is necessary for the individual steps of the analysis in question.
Discussion
4. Given the difference in weight of the participants, together with the small sample size, to what extent are the data obtained generalizable to all categories of the sport?
Yes, it can be because female kickboxers included in the study compete in different weight categories on international level. On international level the is limit for persons from country in weight category. Because of this limit Slovakia and Poland, but also other countries have limited fighters in their representations. But for other research it would be good to compare female fighters on international level from several counties.
Conclusion
5. Based on the previous questions, do the authors believe that the data obtained can be extrapolated to the generality of women practicing this sport at a high level?
The obtained results can serve as a reflection platform for the standardization of criteria that need to be managed from the point of view of confrontation at the international level. From the point of view of the representation of the countries, it would be necessary in the future to supplement the research file with other female competitors who belong to the elite category.
Round 2
Reviewer 1 Report
Dear Authors,
Thank you for trying to attend to some of my comments in your revised manuscript. I note that the changes are very minimal, with decimal points added alongside some basic, short sentences. Your responses to my comments in the letter should be incorporated into the actual manuscript - not just left in this portal. That way, the reader will learn more about the context of Poland and Slovakia.
Please try to work on these additions to make the article more detailed for the outside reader less familiar with the countries, kickboxing, and female kickboxing in particular. Please also check your responses for typos, as some of these have obvious errors, e.g. wako (which should be WAKO). This gives the impression that the changes were rushed without any proofreading.
Many thanks in advance.
Author Response
Dear Reviewer,
Thank you very much for your time and valuable comments, which all have been considered and incorporated. The detailed list of responses is given below. We hope that the modifications and explanation will be acceptable for you.
Yours sincerely,
Pavel Ruzbarsky , Kristina Nema , Marek Kokinda , Łukasz Rydzik , Tadeusz Ambroży
- 1. Semi-colons between sections should be replaced with full stops / periods and "0,05" should be "0.05" (with the decimal point)
A: This has been corrected.
- Line 26: unnecessary capitalisation of "the"
A: This has been corrected.
- What is the average age of a competitive / elite female kickboxer? Is it younger than in other combat sports? The sample ranging from 21 to 25 is quite young.
A: The average competitive age of elite female kicboxers in senior categories was on last world senior championship 25 years. But also in other combat sports the average age of female fighter is lower than man. It can by caused by the end of studies at university and starting work, or pregnancy and starting a family.
- Are there any statistics on the numbers of female kickboxers in Poland and Slovakia? Are they within national governing bodies?
A: Added information in the manuscript.
- How did you access this population in the first place?
A: Two of author (one from Slovakia and one from Poland) prepare kickboxers from our study on competitions and cooperate with national union on physical fitness testing.
- Are any of the authors involved in kickboxing or in the preparation of these athletes for their competitions?
A: Yes, one of author was female kickboxer on international level and has several medals from world cups. And two of authors cooperate with national union and prepare kickboxers from our study on competitions as we mention up.
- Why was this study conducted? Do Slovak and Polish kickboxers follow different strength training routines? Is one of the nations more dominant in the sport than the other one?
A: Added information in the introduction.
- What was the minimum amount of experience required to be included in the sample?
A: Information and detailed inclusion and exclusion criteria added
- Were there any specific ethical measures that you followed / considered.
A: Described in the text.
- The height and weight differences between the Slovak and Polish kickboxers is interesting. Why might this be? Just by chance? Or is there a general difference in body size and mass between the national populations?
A: Added information in the limitations of the study.
Reviewer 2 Report
The authors have not responded to or corrected most of the issues and recommendations raised.
Author Response
Dear Reviewer,
Thank you very much for your time and valuable comments, which all have been considered and incorporated. The detailed list of responses is given below. We hope that the modifications and explanation will be acceptable for you.
Yours sincerely,
Pavel Ruzbarsky , Kristina Nema , Marek Kokinda , Łukasz Rydzik , Tadeusz Ambroży
- How many observers reviewed the recordings to establish the technical and tactical indicators? Were any of them external to the researchers? (line 110).
A: Added information in the text
- Do you believe that using an isometric fatigue method is the most efficient way to measure strength in a sport like kickboxing?
A: The choice of tests was conditioned by previous diagnoses of players found in the literature
- The analysis of the correlation between technical and tactical indicators is very poor. How are these successes or errors distributed over the combat time? What influence does fatigue have on this aspect? A deeper analysis of this area is recommended.
A: Added information in the limitations
- Given the difference in weight of the participants, together with the small sample size, to what extent are the data obtained generalizable to all categories of the sport?
A: This has been corrected.
- Based on the previous questions, do the authors believe that the data obtained can be extrapolated to the generality of women practicing this sport at a high level?
A: This has been corrected.